# Hip Fracture Prevention in Osteoporotic Elderly and Cancer Patients: An On-Line French Survey Evaluating Current Needs

**DOI:** 10.3390/medicina56080397

**Published:** 2020-08-07

**Authors:** Laëtitia Rodrigues, François H. Cornelis, Sylvie Chevret

**Affiliations:** 1INSERM UMR 1153, Equipe ECSTRRA, Service de Biostatistique et Information Médicale, AP-HP Hôpital Saint Louis/Université Paris Diderot, 75010 Paris, France; sylvie.chevret@u-paris.fr; 2Department of Radiology, Tenon Hospital, Sorbonne Université, 4 rue de la Chine, 75020 Paris, France; francois.cornelis@aphp.fr

**Keywords:** hip fracture, proximal femur, osteoporosis, bone metastases, prevention

## Abstract

*Background and objectives*: Hip fracture is a major public health issue. Those fractures lead to high costs and a decrease in quality of life. A national French survey was conducted, with the objectives to firstly assess the current management of hip fracture and its prevention, both in the osteoporotic and cancer settings, and secondly to evaluate the opinions of physicians on the potential use of minimally invasive implantable devices to prevent hip fracture in alternative of surgery. *Materials and methods*: This national survey was conducted in France between April and July 2017. Questionnaires were sent to orthopedic surgeons, interventional radiologists, oncologists, and rheumatologists. Completed questionnaires were analyzed and compared according to two indications: orthopedics-traumatology and oncology. Factors associated with these responses were assessed using univariable analyses, based on chi-square tests or an exact Fisher test, as appropriate. *Results*: A total of 182 questionnaires were completed and further analyzed. Physicians have highlighted the need for a low re-fracture rate and to improve life expectancy for more than 1 year (50% for responders of the orthopedics-traumatology questionnaire and 80% for the responders interested in both indications), as well as quality of life (12.5% and 31%, respectively), but with no significant differences in the oncologic indication. Most of the experts were willing to use or prescribe implantable devices for prevention (63% in orthopedics-traumatology and 93% in oncology), although limited clinical experience (54 and 58%) and surgical risk (around 30% in each indication) were considered as limits. *Conclusions:* Prevention of hip fracture remains a concern for physicians. More clinical experience with implantable devices, in particular in cancer patients, is needed, but implemented in a strategy to maximize patient recovery while reducing costs.

## 1. Introduction

Hip fractures, due to bone fragility, either caused by osteoporosis or the presence of metastases in the bone, can be devastating for elderly and cancer patients [1,2,3,4,5,6]. It reduces quality of life and life expectancy. In the following 5 years, at least 20% of osteoporotic patients presenting with hip fracture will suffer from a contralateral hip fracture (10% at 1 year, 15% at 2 years), with a 15–30% mortality rate [7,8]. A global call for action has been initiated by the Fragility Fracture Network (FFN), involving several leading medical organizations worldwide, to improve the care of patients suffering from hip fracture [9]. The aims of this collaboration are to maximize patient recovery, i.e., to restore function and subsequent fracture prevention using non-invasive and invasive techniques [9,10]. Pharmaceutical companies have promoted therapies that demonstrate little effectiveness in reducing the risk of hip fracture in osteoporotic elderly, as reported by Bawa et al. [11], with a reduction of 30% after the age of 85 years, and very low prescription and patient adherence [12,13]. Moreover, even when the treatment is correctly followed, the earliest onset of benefit for hip fracture prevention is reported after 12–36 months [14,15].

Hip fractures are also commonly observed in cancer patients with bone metastases, the most frequent location being the proximal part of the femur [16]. These pathological fractures can be very disabling when symptomatic and difficult to operate, as this population already has a reduced life expectancy [17]. Therefore, it is crucial to detect and prevent them as early as possible [18]. The use of medication has been suggested to prevent adverse effects of cancer treatments on bone health and osteoporotic patients [19,20]. Standard surgical osteosynthesis can be performed, in particular, in cancer patients with impending fractures, according to the Mirels scoring system [21,22,23,24,25,26]. As alternatives, different interventional options to prevent hip fracture have been suggested [27,28,29], but it remains unclear how such a device would be accepted in the routine.

In this context, a national French survey was conducted, with two main objectives: First, we aimed to assess the current management of hip fracture and its prevention, both in the osteoporotic and cancer settings, and secondly, we aimed to evaluate the opinions of physicians on the use of innovative minimally invasive implantable devices to prevent hip fracture.

## 2. Materials and Methods

### 2.1. Study Design and Selection

A national survey was conducted in France between April and July 2017. The selection of physicians was made according to patients’ standard of care and two main indications: (1) orthopedics-traumatology and (2) oncology.

In orthopedics-traumatology, physicians were those participating in orthogeriatric and/or fracture liaison services involved in osteoporosis treatment, i.e., rheumatologists and gynecologists for screening and prevention of osteoporosis and orthopedic surgeons for fracture repair.

In oncology, physicians were participants of multidisciplinary meetings, i.e., orthopedic surgeons for bone consolidation, interventional radiologists for radiotherapy or other mini-invasive treatments, and oncologists for primary cancer.

Physicians were contacted through medical French Societies in the orthopedics (i.e., SOFCOT (French Society of Orthopedics and Traumatology Surgery), SFHG (French Society of Hip and Knee Surgery)), interventional radiology (i.e., SFR (French Radiology Society), FRI (Interventional Radiology Federation)), oncology (i.e., SFC (French Cancer Society), SFCO (French Society of Oncologic Surgery)), and rheumatology (i.e., SFR (French Society of Rheumatology)) fields.

### 2.2. Questionnaire

The questionnaire content was based on a literature review, clinical experience, and experts’ knowledge regarding hip fracture [29,30]. This online survey collected information on the opinions and experience of practitioners on hip fracture prevention in the case of osteoporosis or metastatic tumors. Three specific themes were addressed: previous knowledge or experience with the studied device (Section 1); profile of physicians interested in hip fracture prevention (Section 2); hip fracture epidemiology, clinical practice in both indications, and physicians’ opinions about surgical prevention and the potential of innovative devices (Section 3). A total of 67 questions were proposed, divided into three sections. All the practitioners had to reply to the first 4 questions about the studied device (Section 1) and then 7 questions about themselves (Section 2). According to their field of expertise, they were then directed either to a set of questions regarding the orthopedics-traumatology indication (30 questions) or the oncology indication (26 questions), or both successively (Section 3) (detailed in the supplementary files, Appendix A: Elicitation questionnaire). Questions included 56 single-choices answers, 11 multiple-choices answers, and no free answers other than for the choice « other » in a question they had to complete. Among devices discussed, Y strut was chosen as an illustrative example of an innovative minimally invasive implantable device [31,32,33], given that it was the most recent device obtaining a CE-mark (European Community authorization to market) and was available in the market at the time of the study elaboration. Y-STRUT^®^ (Hyprevention, France) is a medical device made of radio-transparent polyether ether ketone (PEEK) polymer (biomechanical characteristics of cortical bone) and combined with poly (methyl methacrylate) (PMMA) cement for bone anchoring. It is implanted through a minimally invasive procedure, under imaging control and with a specific instrumentation to safely assemble the two implants in situ into the proximal femur.

### 2.3. Data Collection

Electronic Data Capture (REDCap), a web application [34] hosted at the biostatistics unit of Saint-Louis Hospital (Paris, France), was used to design the questionnaire accessible on the Internet and collect the answers. A reminder was sent once after a few weeks. Responders were not paid and received no reward; all questionnaires were anonymous. Answers were analyzed to check replies and avoid eventual duplicates. This research operated within the framework of the reference methodology for treatments, including health data, carried out as part of research in which the patient gave his/her non-opposition to participate after being individually informed (MR-003).

### 2.4. Statistical Analysis

No formal sample size computation was performed, given that no specific testing was to be done. Nevertheless, we computed that a sample size of 100–200 respondents would allow to estimate any prevalence of interest (ranging from 10 to 90%), with a 95% confidence interval of width from 0.14 to 0.20 [35].

Summary statistics, namely percentages, are reported. Questionnaires were distinguished according to the indication, either orthopedics or oncology.

Questionnaires completed only for the first sections (about the studied device and/or themselves) were excluded, given that no information regarding the potential indications of the device could be analyzed. A total of 203 questionnaires were fulfilled, but 21 (10%) were incomplete, with no reports of information on either orthopedics or oncologic assessments. They were thus excluded from further analyses. A total of 182 questionnaires were reviewed from 127 physicians, namely, 48 with interest in the orthopedics indication only, 24 only in the oncological indication, and 55 in both (Figure 1). It should be noted that, before exclusions, 127 and 28 of the respondents were orthopedic surgeons and interventional radiologists, respectively, and 18 were radiologists, whereas only 11 were rheumatologists and 2 were geriatricians. No oncologists answered this survey.

The characteristics of physicians who completed the questionnaire according to their interest in the indications of the device are described on Table A1, exhibiting no significant difference in respondents except on the specialty and previous information on the device. Expectedly, those interested in only orthopedics indication were more likely orthopedic surgeons (45/48, 94% vs. 9/24, 37.5%, *p* < 0.0001). It should be noted that about one half of those interested in the oncological indication had never heard about the device, compared to 8 out of 10 of those interested in its orthopedics use (*p* = 0.003).

The typical profile of the respondent was a male, age of 45 years or older, orthopedic surgeon, and working either on a public or private facility. Most of the physicians had never heard about the innovative device Y-STRUT^®^ before this survey (78%), and among the physicians knowing about it (22%), only a few had already used or prescribed it (21%).

Factors associated with these responses were assessed using univariable analyses, based on chi-square tests or an exact Fisher test, as appropriate. Analyses were performed using R statistical software version 3.2.0 (R foundation for Statistical Computing, Vienna, Austria, available online at http://www.R-project.org).

All tests were two-sided, with *p*-values of 0.05 or less denoting statistical significance.

## 3. Results

### 3.1. Orthopedics Assessments

Table A2 reports the main features of responses to the questionnaires on the orthopedics indication. Regarding the orthopedics-traumatology indication (*n* = 103, 57%), physicians reported that they could be users of the studied device at 49%, prescribers at 13%, and both at 38%.

Concerning the patients’ profile on their current practice, practitioners agreed that women were in the majority (94.5%) at more than 75 years old (98%). They estimated the number of patients with a hip fracture due to aging and osteoporosis within a year, either per trochanteric or femoral neck fracture, and mainly caused by a simple fall. The history of fractures may also impact the risk profile of these patients (i.e., wrist (40%) or one (34%) or several vertebrae (15%)).

In these patients, osteoporosis is mostly undiagnosed or untreated before the first hip fracture (58%), and thereafter even less (22%) or ignored (40% are not prescribers). For the physicians, the main criterion for choosing the material (mostly a nail, according to the respondents) for fracture reduction was the hip fracture criterion. They estimated a majority (in 54.5%) of contralateral hip fractures between 0–10 per year. Regarding prevention of contralateral hip fracture, only 4% suggested no prevention. Prevention with such an innovative device is considered to be applied simultaneously with the treatment of the first hip fracture (36%) or eventually postponed (30%) at the 3-month visit (44%) or after 6 months (19%). The impact of contralateral fractures was acknowledged, both in terms of place of life or life expectancy: Patients after contralateral hip fracture are mainly discharged to a healthcare facility (35.4% as compared to 4.2% after the first fracture), while life expectancy is estimated to be less than 12 months in 47% of these patients (with 24.5% < 6 months), as compared to the previous 22%. Finally, physicians did not really agree about the fracture risk reduction at 1 year, which may convince them to use such an innovative device: 30% of risk reduction at 1 year for 15% of the responders, 50% of risk reduction at 1 year for 31.5% of the responders, 70% of risk reduction at 1 year for 18.5% of the responders, and 90% of risk reduction at 1 year for 4% of the responders (for the majority, it is a risky reduction when equal or superior to 50%).

### 3.2. Oncology Assessments

For the oncology indication (*n* = 79, 43%), similarly, physicians reported that they could be users of the studied device at 54%, prescribers at 17%, and both at 30%. Table A3 reports the main features of responses to the questionnaires on the oncology indication.

In their current practice, they reported that the number of patients with tumorous pre-fractural lesions at the proximal femur was less than 5 per month (93%). There is not a strict majority concerning the type of lesion, but femoral neck seems the most common (43%) before per trochanteric (22%) and all locations equally (27%) (i.e., femoral head (2%), diaphysis (6%)). These types of lesions cause less than 10 hip fractures per year (82%). The majority of practitioners also agreed on the type of primary cancer of these patients suffering from bone metastasis, at the levels of the proximal femur, breast (76%), lung (51%), prostate (44%), and kidney (33%), being the most common. In these patients, a medical treatment for bone metastasis at the level of the proximal femur was mainly prescribed (79%). Most of the specialists have declared to propose a preventive treatment against proximal femur fracture mainly regarding the patients’ pain (45%) and Mirels’ score (28%). Patients are sent either to the orthopedics unit (35%) or to a multidisciplinary meeting (30%). For the physicians, the main criteria for choosing the material used are the extent of the lesion, pain, and ease-of-use. In case of bone metastasis, the main preventive treatment proposed was osteo-synthesis at the orthopedics operative room, though 16% does not do any prevention. Surgical risk was the main reported risk to not use/prescribe such an innovative device in prevention, whatever the indications (34.5% vs. 8% for oncology only, *p* < 0.05), suggesting that interventional radiologists are rather confident on this operative technique.

Finally, most of the experts would be willing to use or prescribe an innovative device in prevention in both indications and especially for cancer patients (63% in orthopedics-traumatology, 93% in oncology, *p* < 0.05). Among the benefits for using an innovative device, such as Y-STRUT^®^, the doctors highlighted the need for a low re-fracture rate and high life expectancy at 1 year, as well as quality of life, but they also agreed on the major risks of such device (more than 50% in each indication: 54% orthopedics-traumatology, 58% oncology), which include the limited clinical experience and the associated surgical risk (around 30% in each indication). It should be noted that, for the benefits, the re-fracture rate at 1 year was significantly different between responders of the orthopedics questionnaire compared and the responders interested in both indications (50% vs. 80%), as well as for life expectancy at 1 year (35% vs. 58%), patient dependency (12.5% vs. 31%), and no benefit/no use (21% vs. 0%); however, there were no significant differences in the oncologic indication for the benefits. Only the surgical risk was significantly different between responders of the orthopedics questionnaire compared to both indications (48% vs. 29%) and between responders of the oncologic questionnaire compared to both indications (8% vs. 34.5%) (Figure 2).

## 4. Discussion

This French national survey regarding physicians’ opinions confirms that hip fracture prevention remains a challenge and may need to develop new approaches, in particular for cancer patients. However, the strategy remains not clearly defined as several specialties were involved. Indeed, physicians who answered this survey are mainly orthopedic surgeons, who are involved in fracture consolidation, once the event occurred. Interventional Radiologist are involved in the prevention of impeding fractures. Rheumatologists and geriatricians (as well as gynecologists) are more involved in osteoporosis diagnosis and prescription of preventive medications. Unfortunately, no information was obtained from oncologists who are following the patients. However, this survey highlights the urge to develop multidisciplinary collaboration, such as ortho-geriatric services (as well with rheumatologists) or fracture liaison services [5,36,37], and to create more dialog between physicians to correctly screen and prevent hip fractures.

In osteoporotic patients, surgical prevention of contralateral hip fractures is still debated, as contralateral fracture concerns 9% of patients with a recent hip fracture at 1 year up to 20% at 5 years. Furthermore, there is a very low prescription of drug treatment, partly due to low osteoporosis diagnosis before fracture (60% of physicians answered no), no automatic prescription after the first fracture, and lack of adherence for these high-risk patients [12,13,29,38]. The efficacy of these treatments is not clearly demonstrated, with at best 50%, and delayed at up to 12–18 months [7]. Physicians answered that 35% of patients are mainly discharged to a healthcare facility after contralateral hip fracture, compared to only 4% of patients after the first hip fracture. One out of three elderly patients return to their previous level of dependence after a hip fracture, while 50–60% cannot walk alone and need assistance in daily living activities and 25% need full-time nursing-home care [36,39]. Therefore, almost all of the physicians agreed on surgical prevention (95%), either simultaneously to the treatment of the first hip fracture or postponed up to 3–6 months, to provide a complement to the pharmaceutical approach. Surgical solutions have the advantages to be immediate and durable but may also raise ethical and medical issues [4,5,6,7] considering that the bone is not yet fractured and there are risks linked to surgery in frail patients. Currently, innovative devices are implanted through a minimal invasive approach but may still need to be accepted through more clinical experience [31,32,33] to show a positive benefit-to-risk ratio of surgery in such a case. Indeed, it was confirmed by the French experts, who seemed mostly willing to use or prescribe an innovative device; however, a lot remained hesitant, mainly due to the limited clinical experience (55%) and the associated surgical risks (29%).

In oncology, patients with metastatic disease are often fragile and have a limited life expectancy [16]. The usual treatment described by the French experts is rather classic with chemotherapy, radiotherapy, or both, when needed. French physicians added that a fracture preventive treatment is often prescribed (63%), the most common being a standard osteosynthesis in the orthopedics unit (64%). Indeed, it has been shown that there is a higher survival rate with prophylactic fixation of metastatic femoral lesions before pathological fractures, combined with a relatively low perioperative risk compared to standard fracture treatment [17,18,21]. Several minimal invasive prophylactic techniques are under investigation but are not standardized and therefore are hardly reproductible [22,23,25]. These results suggest that such innovative devices are less invasive and when used as prevention, may be sufficient and safe solutions for these frail patients, often recused from standard surgery [31]. Additionally, French physicians seemed mostly willing to use or prescribe such an innovative device in oncology, but again, a lot remained hesitant (58%), mainly due to a limited clinical experience.

This study has some limitations. Although health surveys are important sources of information for evidence-based medicine, biases in questionnaires could be an issue. Thus, we carefully checked the question wording to avoid ambiguous or lengthy questions, as well as technical jargon. Given the fact that we collected data regarding beliefs and behaviors, one cannot exclude some reporting biases. One cannot exclude that some selection bias occurred, given that the questionnaire was distributed via professional societies and those who responded were probably not representative of the total physicians’ target group, with physicians particularly interested in devices that are overrepresented among the respondents. Notably, although the questionnaire was sent to oncologists, more answers from orthopedic surgeons and interventional radiologists, that is, those using and not only prescribing the medical devices, were obtained. Questionnaires may have been somewhat too long, so that one cannot exclude induced fatigue among respondents who resulted in inaccurate answers. Only French physicians participated in this survey, suggesting that these conclusions might not be generalized to other countries, even if the literature seems in agreement with the results found in this study. The statistical power is also limited, as only 186 questionnaires were fully completed, though we computed that a sample size ranging from 100 to 200 would allow us to estimate any prevalence of interest (ranging from 0.1 to 0.9) with a width of 95% confidence interval below 0.20, which appeared reasonable [35]. It should also be noted that some physicians preferred to not give their opinion regarding both indications, with about 35% of missing values in each indication. The total number of sent questionnaires remains unknown as several scientific societies just gave one email for the whole diffusion list and no acknowledgment of receipt was received.

## 5. Conclusions

This online survey reported French physicians’ current practice of prevention of hip fractures. French physicians who responded to the questionnaires who were possibly particularly interested but with a somewhat limited clinical experience seemed mostly willing to use or prescribe innovative devices for prevention of fractures in both indications. This study helps us to understand why multidisciplinary evaluation is necessary to help the suffering of these patients from osteoporosis or cancer and to provide them better care at lower costs. It also confirms that pursuing investigations to get more clinical experience of interventional techniques is needed to help physicians in their practice. Thus, this study may be very informative for either practitioners or industrials developing such innovative approaches.

## Figures and Tables

**Figure 1 medicina-56-00397-f001:**
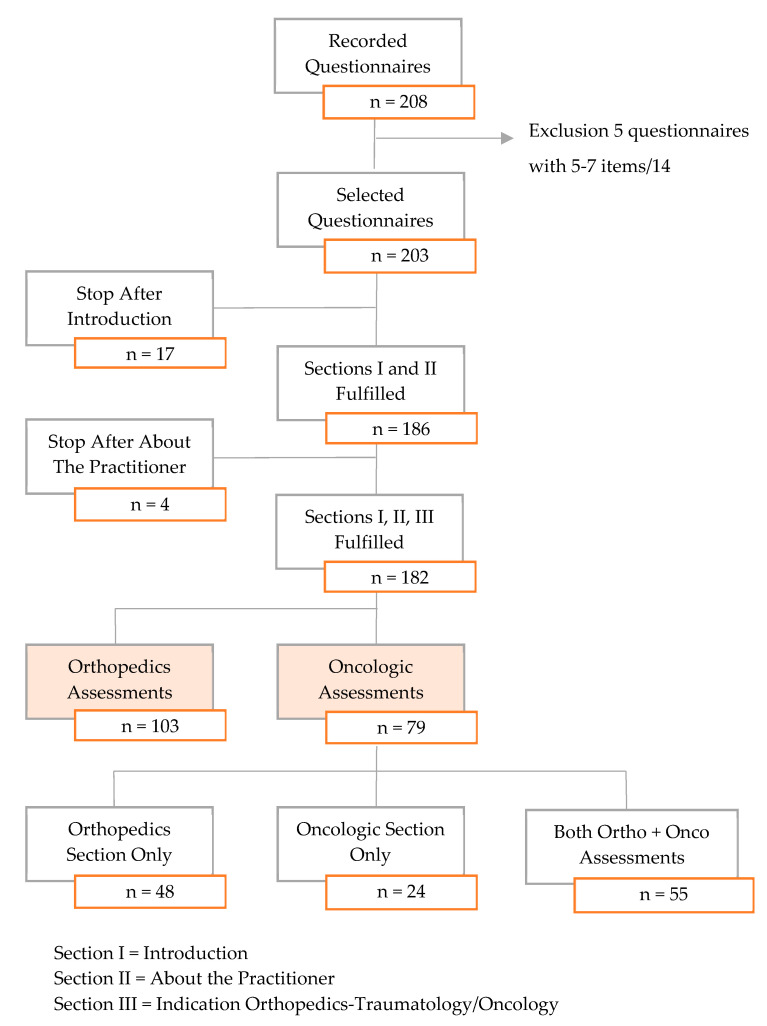
Flowchart of the physicians who completed the questionnaire.

**Figure 2 medicina-56-00397-f002:**
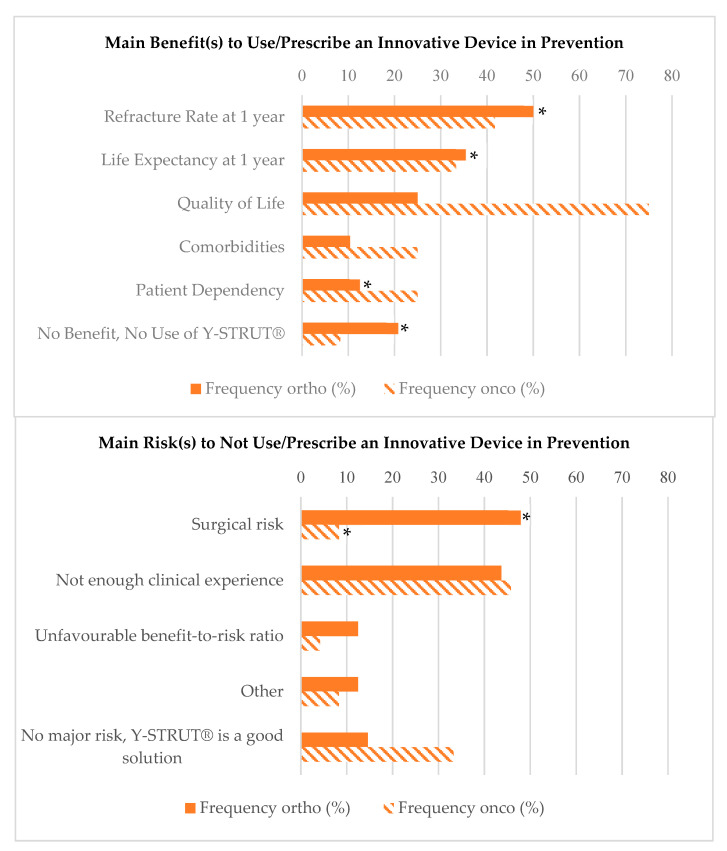
Reported benefits and risks for using an innovative device in prevention, such as Y-STRUT^®^, according to the indication. * *p*-value <0.05 between ortho/onco only and both indications.

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
