# Peer review of "Hip Fracture Prevention in Osteoporotic Elderly and Cancer Patients: An On-Line French Survey Evaluating Current Needs"

_medicina, 2020, doi:10.3390/medicina56080397_

Round 1

Reviewer 1 Report

Hip fractures are one of the most common health issues.

Rodrigues et al. submitted a questionnaire to physicians on the potential use of minimally invasive implantable devices to prevent hip fracture in alternative of surgery and analyzed 182 questionnaires.

They underline mostly the need of life expectancy and life quality improving. Moreover, they indicate that most of the interviewed were disposed to use or prescribe implantable device for prevention of fractures, but they pointed out the limited clinical experience of physicians and surgical limits.

  • In the introduction, it is not clear why and how do the authors want to relate hip fractures to osteoporosis and cancer.
  • Furthermore, references related to fracture incidence and treatments are not recent.
  • The introduction is poor to understand the aim of the study, please improve it. 
  • If the questionnaire included oncology assessment, why did not the authors include oncologists in their study?
  • Which could be the concrete applications of the study? The authors should suggest in the conclusions section.

Reviewer 2 Report

The authors have presented an interesting manuscript on the results of a survey of health professionals on the prevention of hip fractures in cancer patients. However, the study cannot be published in its current version.
The abstract and introduction sections are well formulated. In the material and methods section, it is necessary to indicate how many professionals the survey was sent to, how many responded, the power of the study and, most importantly, whether the sample studied is representative of the total population.
Statistical analysis subsection. The reviewer advises the authors to explain why "questionnaires completed only for the first sections were excluded".
Results section.
Subsection Characteristics of practicioners included in this study.
The reviewer advises the authors to move the first paragraph of this section and figure 1 (flowchart) to the methods section.
Discussion section.
Subsection limitations.
The reviewer advises the authors to expand this section with all biases and their consequences. It is okay to cite the power of the study in this subsection, but it is correct to explain your calculation in the methods section.
Conclusions subsection
The reviewer finds that the conclusions are very generic and that the authors should further refine the conclusions based on the study results.
